# Harnessing the open access version of ChatGPT for enhanced clinical opinions

Zachary M. Tenner[1]*, Michael C. Cottone[1], Martin R. Chavez[1,2]

1 New York University Grossman Long Island School of Medicine, Mineola, New York, United States of America, 2 Department of Obstetrics and Gynecology, New York University Langone Health–Long Island, Mineola, New York, United States of America

* Zachary.Tenner@nyulangone.org

## Abstract

With the advent of Large Language Models (LLMs) like ChatGPT, the integration of Generative Artificial Intelligence (GAI) into clinical medicine is becoming increasingly feasible. This study aimed to evaluate the ability of the freely available ChatGPT-3.5 to generate complex differential diagnoses, comparing its output to case records of the Massachusetts General Hospital published in the New England Journal of Medicine (NEJM). Forty case records were presented to ChatGPT-3.5, prompting it to provide a differential diagnosis and then narrow it down to the most likely diagnosis. The results indicated that the final diagnosis was included in ChatGPT-3.5's original differential list in 42.5% of the cases. After narrowing, ChatGPT correctly determined the final diagnosis in 27.5% of the cases, demonstrating a decrease in accuracy compared to previous studies using common chief complaints. These findings emphasize the necessity for further investigation into the capabilities and limitations of LLMs in clinical scenarios while highlighting the potential role of GAI as an augmented clinical opinion. Anticipating the growth and enhancement of GAI tools like ChatGPT, physicians and other healthcare workers will likely find increasing support in generating differential diagnoses. However, continued exploration and regulation are essential to ensure the safe and effective integration of GAI into healthcare practice. Future studies may seek to compare newer versions of ChatGPT or investigate patient outcomes with physicians integrating this GAI technology. Understanding and expanding GAI's capabilities, particularly in differential diagnosis, may foster innovation and provide additional resources, especially in underserved areas in the medical field.

## Author summary

Integrating artificial intelligence (AI) into clinical medicine has long been a technological goal. Since its release in November 2022, ChatGPT has gained popularity, sparking questions about its proficiency in enhancing patient care. AI has demonstrated its ability to answer multiple-choice questions and exams at a level equivalent to medical students. It also excels in scenarios involving common chief complaints. However, ChatGPT's ability to participate in advanced clinical conversations and provide difficult patient diagnoses

**Data Availability Statement:** All relevant data are within the manuscript.

**Funding:** The author(s) received no specific funding for this work.

**Competing interests:** The authors have declared that no competing interests exist.

has been largely unexplored. In this study, we investigated the capability of ChatGPT-3.5 to generate complex differential diagnoses by presenting 40 clinical case reports sourced from the New England Journal of Medicine (NEJM). Overall, ChatGPT-3.5 accurately identified the correct differential diagnosis 27.5% of the time. As we transition towards a medical landscape where physicians may leverage AI as a clinical tool, this study emphasizes both the limitations and potential of ChatGPT. We underscore the ongoing need to define AI capabilities to ensure its safe integration into medical practice and advocate for the sustained open accessibility of generative AI for patient care.

## Introduction

Since the 20<sup>th</sup> century, research and speculation regarding the integration of artificial intelligence (AI) into physician reasoning has been ongoing. In 1987, Schwartz et al. asserted that "major intellectual and technical problems must be solved before we can produce truly reliable [healthcare] consulting programs" [1]. While models of clinical problem-solving have been described for years, it is only recently that technology has advanced sufficiently to investigate the role of AI in clinical medicine. OpenAI's ChatGPT (Generative Pre-trained Transformer), one of the world's first widely used Large Language Models (LLM), uses billions of parameters to generate user-informed text. In the healthcare sector, this Generative Artificial Intelligence, (GAI) encompasses a wide range of medical knowledge that can be tailored to the user's needs, from assisting medical students with United States Medical Licensing Exam (USMLE) questions to creating next-generation sequencing reports with treatments options for oncologists [2,3]. Since its release, professionals began assessing the value of ChatGPT by pushing its limits within medical knowledge; however, it is imperative to explore ChatGPT's role in patient care to best demonstrate and provide direction for how health professionals will work with AI as technology continues to develop [4,5].

ChatGPT has distinguished itself by achieving passing scores on the USMLE examination, equivalent to those of a third-year medical student [2]. This accomplishment opens the gates for potential applications of the model in medical education, serving as an interactive tool for medical school and an overall support for clinical thinking. Radiology and pathology have received significant attention in GAI research, with efforts focused on enhancing LLMs to better understand images and detect cancers. Despite receiving no specific training in either subject, "ChatGPT nearly passed a radiology board-style examination without images," and demonstrated accuracy in "[solving] higher-order reasoning questions in pathology" [6,7]. Ali et. al. identified ChatGPT's ability to perform at high rates on the neurosurgery oral boards examination while emphasizing the limitation of using multiple-choice examinations to assess a neurosurgeon's expertise in patient care [8]. Although ChatGPT has proven effective in choosing from a list of options, the role of LLMs in clinical management has been highlighted as area requiring further research.

Mirroring the progression of a medical student, the next logical step is to evaluate the chatbot's ability to come up with differential diagnosis. These are fundamental to clinical medicine, and the proficiency of ChatGPT in producing medically rational differential diagnoses remains largely unexplored. Hirosawa et al. determined that ChatGPT can successfully create comprehensive diagnosis lists for common chief complaints [9]. Additionally, Rao et al. assessed ChatGPT's ability to generate differential diagnosis for issues routinely encountered in healthcare settings and found that "the LLM demonstrated the highest performance in making a final diagnosis with an accuracy of 76.9%" [10]. Previous research has done an excellent job of

assessing ChatGPT's ability to pass multiple-choice exams and provide differential diagnosis for standard chief complains with high accuracy; however, the generalizability of ChatGPT to more complex clinical scenarios must be examined [11].

To comprehensively assess the potential of GAI and LLMs in complex medical reasoning, we conducted a study to evaluate the ability of the freely available ChatGPT-3.5 to provide differentials on case records of the Massachusetts General Hospital published in the New England Journal of Medicine (NEJM). Our research takes a unique approach by utilizing clinical case reports identified by the journal to establish novel medical or biological understanding, thereby further evaluating the chatbot's language capabilities. At the time our study was done, ChatGPT-3.5 has a knowledge cutoff date of September 2021. Consequently, we examined ChatGPT's ability to use clinical reasoning to diagnose 2022 case reports, avoiding reliance on its search function to locate published articles. The primary aim of this study is to evaluate the proficiency of the freely available ChatGPT-3.5 in generating complex differential diagnoses. We intend to compare the chatbot's complete diagnosis list and final diagnosis against the published differential diagnosis for the NEJM case reports. Our hypothesis posits that the percentage of differential diagnoses generated by ChatGPT-3.5 will match the NEJM final diagnosis for the case reports approximately 50% of the time. By elucidating ChatGPT's potential in offering differential diagnoses, we propose future clinical problem-solving cases consider utilizing GAI as an augmented clinical opinion.

## Methods

We presented forty case records from the Massachusetts General Hospital, as published in the New England Journal of Medicine (NEJM) in 2022, to ChatGPT-3.5. All text preceding the "Differential Diagnosis" headline was included, excluding figures. ChatGPT was initially prompted with the instruction, "Provide a differential diagnosis from the following clinical case." Following the generation of a complete list of differential diagnosis, we further inquired, "Can you narrow down the differential to the most likely diagnosis?" Subsequently, we recorded whether the final diagnosis, as referenced in the NEJM, was included in ChatGPT's complete differential diagnosis list. Additionally, we noted whether ChatGPT's "most likely diagnosis" aligned with the final diagnosis noted in NEJM.

## Results

Among the 40 cases presented to ChatGPT-3.5, 23 cases (57.5%) were not considered in its original differential list. The average length of the original differential list generated by ChatGPT was 7±2 possible diagnoses, ranging from a high of 12 and a low of 3. The length of the differential appeared random. In 17 cases (42.5%), ChatGPT did include the final diagnosis in its original differential list. After narrowing down its differential list, ChatGPT correctly identified the final diagnosis in 11 cases (27.5%) and eliminated the correct diagnosis in 6 cases (15%). These results are presented in Fig 1.

## Discussion

The role of Generative AI (GAI) and Large Language Models (LLMs) in clinical medicine is a rapidly growing area of research. Assessing the potential and limitations of ChatGPT (v3.5) within the scope of patient care is essential to determine how and where it can best be utilized. We decided to focus on the complimentary version of ChatGPT to ensure the largest possible audience has access to this technology. Presenting 40 case records from the New England Journal of Medicine (NEJM) to ChatGPT allowed us to delve deeper into the role of LLMs in healthcare, specifically studying their success rates in producing differential diagnoses of

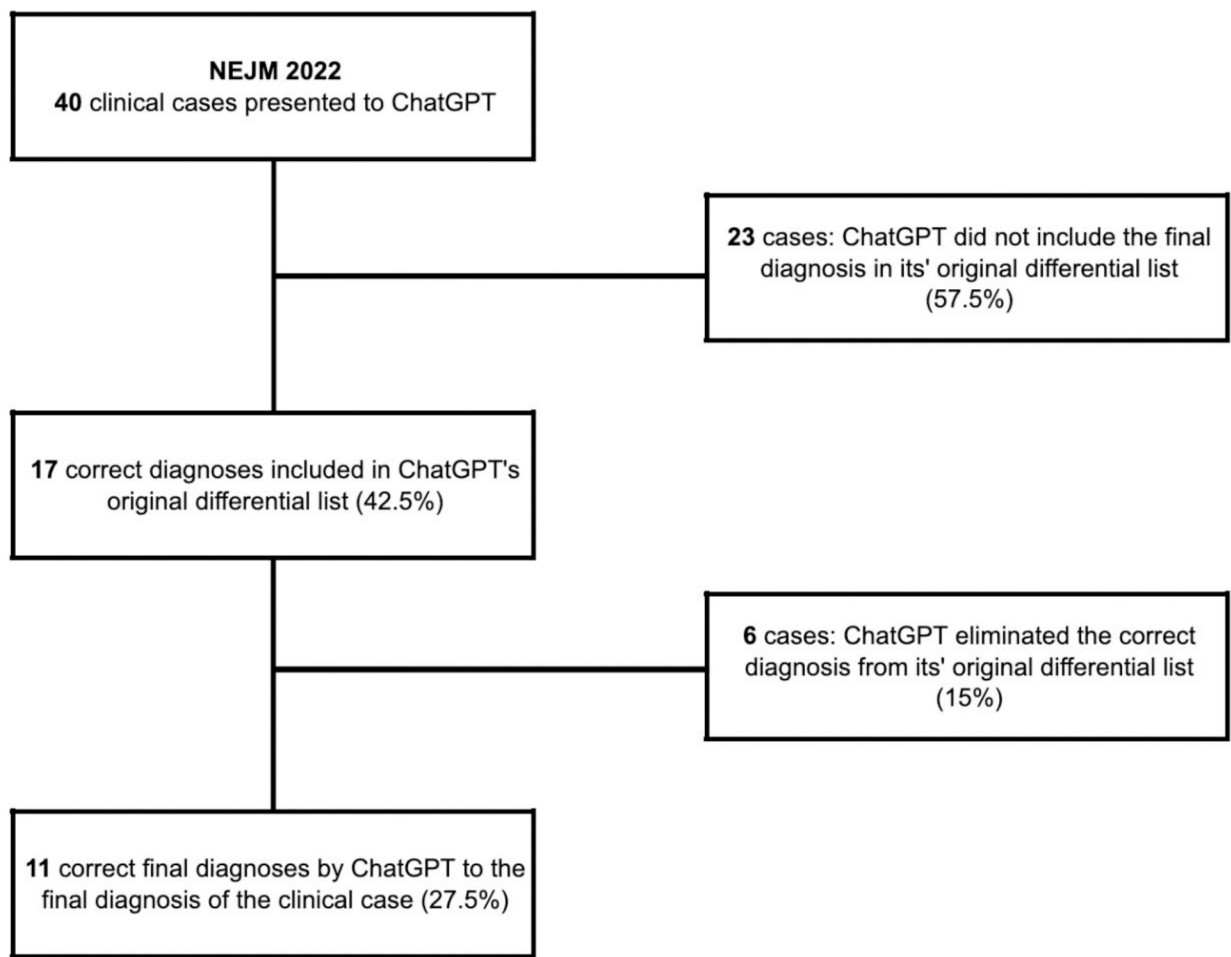

**Fig 1. Flowchart of the 40 case records of the Massachusetts General Hospital that were published in the NEJM after being presented to ChatGPT.**

complex patient presentations. ChatGPT accurately identified the correct differential diagnosis 27.5% of the time. Notably, the differential list accuracy of ChatGPT, when presented with clinical vignettes of common chief complaints, has been reported to be over 80% [9]. However, this accuracy dropped by over 50% when we increased the difficulty from common chief complaints to complex clinical cases using NEJM case reports.

Furthermore, our results can be compared to Kanjee et. al., where the authors utilized NEJM clinicopathologic conferences as challenging medical cases. Their assessment of Chat GPT-4 "provided the correct diagnosis in its differential in 64% of challenging cases and as its top diagnosis in 39%." Of note, our assessment of the open access version of ChatGPT was about 20% lower when including the diagnosis in its differential and 12% lower when selecting the final diagnosis [12]. GPT-4 was also compared to medical-journal readers to assess its ability solve complex clinical cases, as it correctly diagnosed 57% of cases [13]. Our research presented diagnosis percentages that were slightly below Kanjee and Eriksen, sparking a conversation about the clinical abilities of GPT-3.5 vs GPT-4. The open-access aspect of GPT-3.5 remains important to research, encouraging its usage among the medical community without a financial investment in GAI. Establishing baseline limitations of ChatGPT allows for future comparisons of its growth and development and ensures cautious use in patient care.

Furthermore, it can provide insight to how to best adjust ChatGPT's settings to better identify the categories for which it receives the highest score.

In the imminent future, physicians and other healthcare workers will likely practice in a world where the latest research journals and electronic medical records are directly linked to Chat-like software. With these upcoming additions to GAI, we anticipate its continued growth in developing differential diagnoses. Therefore, it becomes increasingly important for the field of medicine to better comprehend this information. In both primary care and specialty settings, GAI provides a new medium for physicians to cultivate new ideas, consider novel diagnoses, and consult with a "colleague" when one may not be readily available, especially in rural settings [14].

Future studies may look to expand from our baseline findings. For example, the newer versions of ChatGPT-3.5 do not have knowledge cutoff date and are instead able to pull up-to-date information from the internet. How do newer versions of ChatGPT compare to ChatGPT-3.5? Do patients experience improved outcomes when their physicians integrate ChatGPT into their care? These questions, among others, require elucidation through further experimentation. However, before ChatGPT becomes a new tool within a physician's practice, it is imperative to continue defining and describing its abilities to ensure safe and appropriate reliance on GAI. We strongly advocate for technology companies to consistently offer complimentary versions of generative artificial intelligence. Such accessibility not only maximizes its utilization but also fosters innovation, particularly in the field of medicine.

## Author Contributions

**Conceptualization:** Zachary M. Tenner.

**Data curation:** Zachary M. Tenner, Michael C. Cottone.

**Formal analysis:** Michael C. Cottone.

**Investigation:** Zachary M. Tenner.

**Methodology:** Zachary M. Tenner, Michael C. Cottone.

**Project administration:** Zachary M. Tenner.

**Supervision:** Martin R. Chavez.

**Visualization:** Michael C. Cottone.

**Writing – original draft:** Zachary M. Tenner.

**Writing – review & editing:** Martin R. Chavez.

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
