## [Decision Letter · Decision Letter 0]

12 Dec 2023

PDIG-D-23-00320

Harnessing the Open Access Version of ChatGPT for Enhanced Clinical Opinions

PLOS Digital Health

Dear Dr. Tenner,

Thank you for submitting your manuscript to PLOS Digital Health. After careful consideration, we feel that it has merit but does not fully meet PLOS Digital Health's publication criteria as it currently stands. Therefore, we invite you to submit a revised version of the manuscript that addresses the points raised during the review process.

Please submit your revised manuscript within 30 days Jan 11 2024 11:59PM. If you will need more time than this to complete your revisions, please reply to this message or contact the journal office at digitalhealth@plos.org. Please include the following items when submitting your revised manuscript:

We look forward to receiving your revised manuscript.

Kind regards,

Jennifer N Avari Silva, MD

Section Editor

PLOS Digital Health

Journal Requirements:

2. We ask that a manuscript source file is provided at Revision. Please upload your manuscript file as a .doc, .docx, .rtf or .tex.

3. Please provide separate figure files in .tif or .eps format only and remove any figures embedded in your manuscript file. Please also ensure that all files are under our size limit of 10MB.

Additional Editor Comments (if provided):

Reviewers' comments:

Reviewer's Responses to Questions

**Comments to the Author**

1. Does this manuscript meet PLOS Digital Health’s publication criteria? Is the manuscript technically sound, and do the data support the conclusions? The manuscript must describe methodologically and ethically rigorous research with conclusions that are appropriately drawn based on the data presented.

Reviewer #1: Partly

2. Has the statistical analysis been performed appropriately and rigorously?

Reviewer #1: Yes

3. Have the authors made all data underlying the findings in their manuscript fully available (please refer to the Data Availability Statement at the start of the manuscript PDF file)?

Reviewer #1: Yes

4. Is the manuscript presented in an intelligible fashion and written in standard English?

Reviewer #1: Yes

5. Review Comments to the Author

Reviewer #1: Well-Written Introduction:

The introduction highlights the idea of the entire paper. It raises interest in learning more.

Conceptually Correct:

The authors presented enough mathematical evidence and reasons to support their conceptual correctness.

Overall Presentation:

The overall presentation of the paper, which includes figures, tables, and references, is good. However, a minor revision will be necessary for the final approval of this paper.

Weakness of the Paper

Grammatical mistakes: 

The paper has around 20 grammatical mistakes and several sentence formation problems. A careful revision is mandatory to correct these mistakes. 

No Comparison:

The authors did not compare their results with any of the papers mentioned in the literature review. It is beyond the scope to validate the methodology's effectiveness without proper comparison.

6. PLOS authors have the option to publish the peer review history of their article (what does this mean?). If published, this will include your full peer review and any attached files.

**Do you want your identity to be public for this peer review?** For information about this choice, including consent withdrawal, please see our Privacy Policy.

Reviewer #1: Yes: Sandeep Trivedi

---

## [Editor Report · Decision Letter 1]

11 Jan 2024

Harnessing the Open Access Version of ChatGPT for Enhanced Clinical Opinions

PDIG-D-23-00320R1

Dear Mr. Tenner,

We are pleased to inform you that your manuscript 'Harnessing the Open Access Version of ChatGPT for Enhanced Clinical Opinions' has been provisionally accepted for publication in PLOS Digital Health.

Best regards,

Jennifer N Avari Silva, MD

Section Editor

PLOS Digital Health

Thank you for your responsiveness to previous review.